# Influence of Phosphoric Acid on the Adhesion Strength between Rusted Steel and Epoxy Coating

**Yang Li [1],\*, Bing Lei [2] and Xuqiang Guo [1]**

[1] College of Engineering, China University of Petroleum-Beijing at Karamay, 355 Anding Road, Karamay 834000, China; guoxq@cup.edu.cn

[2] School of Chemical Engineering and Technology, Sun Yat-sen University, Zhuhai 519082, China; leibing@mail.sysu.edu.cn

\* Correspondence: lyanga@cupk.edu.cn

**Abstract:** In this work, we evaluated the influence of phosphoric acid in conjunction with a tannic converter formula on the adhesion of the coating system by employing the epoxy coating as the top coating. The morphology and the composition of the rusted steel before and after treatment were determined by scanning electron microscopy (SEM) and Fourier transform infrared (FTIR) spectroscopy. The adhesion test result showed that the adhesion strength between the rust and the epoxy coating was enhanced from 1.93 MPa to 11.63 MPa with the treatment by 50 g/L phosphoric acid in conjunction with a tannic converter formula. Further investigation of the working mechanism of such treatment revealed that the adhesion improvement could be attributed to the formation of a micro-cracked and compact conversion layer. This layer could reinforce the anchoring of epoxy coating through the pinning effect. It was also found that both the pH and the phosphate radical were the important factors to improve the adhesion strength.

**Keywords:** rust; adhesion; phosphoric acid; tannic acid

## 1. Introduction

It has been known for a long time that the corrosion protection performance of coatings depends mainly on the adhesion of the coating to the substrate [1–3]. It is widely recognized that the application of coatings on rusted steel surfaces is a speculative venture because of the existence of the rust layer, which can deteriorate the interface adhesion [4]. Proper surface preparation is the vital process to obtain the perfect surface state for further coating. Among various surface cleaning techniques, sandblasting is the most effective in terms of oxide removal and provides the surface with suitable roughness for subsequent anticorrosive coatings. However, the application of this technique is limited because of its high cost and operation difficulties in practice. For instance, the location of the equipment, its geometry, and worker's health commitment all needs to be considered in reality [5,6].

As an attractive alternative to sandblasting, the so-called "rust converter" presents advanced ability to clean surface [7]. The "rust converter" can react with the iron oxides that are difficult to completely remove, to provide a proper surface for painting systems [8]. Therefore, the application of rust converter is a world-wide trend in anticorrosive technology [9,10]. Among a variety of rust converters, it is well known that the compositions of the most common rust converters are based on tannic and phosphoric acids [11–14]. It is reported that rust converters based on tannic and phosphoric acids can transform the rust components like lepidocrocite and magnetite as well as goethite into more stable, inert and adherent products [15–22]. Phosphoric acid has long been used as acid pretreatment for corroded steel surfaces and can increase the conversion of the rust efficiently [23,24]. Several authors [25–27] also showed that tannins were thought to be more effective when used in conjunction with phosphoric acid. In addition, according to Ross et al. [16], the adherent product transformed by tannic and phosphoric acid presented small cracks between

the grains. The coating apparently penetrated into the inner rust through the cracks and bound them together. Consequently, the adhesion of the coating system was improved. Furthermore, Gust et al. [17] demonstrated that this transformation of the rust was accompanied with the increasing compactness of the converted rust, which could enhance the cohesion and protectiveness of the rust layer. This cohesion improvement of the rust layer benefit for increasing the adhesion strength of the coating system.

A rust converter treatment based on tannic acid which improved the adhesion between epoxy coating and rusted structural steel has been optimized in our group, but it was not a satisfactory product, since the adhesion strength was not high enough. On the other hand, it was also found that the remainder rust limited the further adhesion improvement of the coating system [28].

For obtaining an optimized rust converter formula which can be employed in industry, the objectives of this paper are: (i) to study the influence of different concentrations of phosphoric acid in conjunction with a tannic rust converter formula on the adhesion strength between rusted steel and epoxy coating and obtaining an optimized concentration of phosphoric acid; (ii) to investigate the influence of phosphoric acid on the composition and morphology of the conversion layer; (iii) to analyze the enhancement mechanism of the adhesion strength.

## 2. Materials and Methods

### 2.1. Materials

The structural steel (Q235), a widely-used constructional material, was selected in the present investigation and its composition was listed in Table 1. The steel was cut into specimens with dimension of 25 mm × 25 mm × 3 mm.

**Table 1.** Chemical composition of structural steel Q235.

| Element | C | S | P | Mn | Si | Cu | Fe |
|---|---|---|---|---|---|---|---|
| wt.% | 0.176 | 0.023 | 0.019 | 0.570 | 0.233 | 0.033 | Balance |

Based on the actual rust formation mechanism, the first blast and then a salt spray method were used to simulate formation of the rust in the laboratory. The specimens were sandblasted to white metal (A Sa2.5) according to ISO 8501-1: 2007 [29], and then exposed for 72 h in the salt spray chamber following the ASTM B117 [30]. Then the rusted samples were dried in the air for 24 h. The tannic acid converter solution was brushed on the rusted samples in one layer after removing the loose rust with soft brush. The samples were allowed to dry in air for 6 h, considering the feasibility in the industrial practice.

An epoxy coating, which has been widely used because of its superior long-term corrosion resistance, was selected as the top coating in the investigation. The epoxy coating was prepared consisting of E-44 resin as binder, amine as hardening agent and dimethylbenzene as solvent, mixed in the stoichiometric proportions of 1:0.8:0.32 and stirred by a magnetic stirrer machine and kept idle for 1.5 h to aging. Since the thickness of the coating has a great influence on the adhesion strength, in order to ensure the thickness, a layer of the prepared epoxy coating was brushed on the substrate and cured in an oven for 24 h at 60 °C. And the average coating thickness measured by a hand-held electronic gage (PosiTector 6000 from Defelsko Corporation, New York, NY, USA according to ISO 2808 [31] was 150 ± 10 μm.e composition of the rust converter employed is as follows (units in g/L): phosphoric acid 5-200, tannic acid 15, sodium molybdate 4, citric acid 1, isopropyl alcohol 10, tertbutyl alcohol 10, and distilled water as the solution. All chemicals and reagents used were of analytical grade.



*2.2. Adhesion Test*

A portable pull-off test (Positest AT-A from Defelsko Corporation, New York, NY, USA) was used to detect the adhesion of the coating system according to ISO 4624-2002 [32]. The diameter of the test dolly was 20 mm with a test area of 3.14 cm$^2$. Two-component glue was applied to fix the test dolly onto the epoxy coating. In order to achieve better adhesion between the surfaces of the test dolly and the epoxy coating, the contact surfaces of the dolly and the epoxy coating were polished with 400 # SiC paper before the adhesion test. After cleaning the surfaces with ethanol, the glue was applied and allowed to dry for 24 h at room temperature. To avoid the adhesion of the epoxy coating on the outside of the test range, the epoxy coating was scoured through to the substrate around the test dolly with a circular hole cutter before running the test [33]. Additionally, six parallel samples were tested and the final result was presented by the average data, and the standard deviation of each result was analyzed by statistical method.

*2.3. Characterization of Rust and Tannic Conversion Layer*

A Magna-IR 560 infrared spectrophotometer (Thermo Fisher, Waltham, MA, USA) was used to measure the IR spectra in the range of 400 to 2000 cm$^{-1}$ with an accuracy of 4 cm$^{-1}$ and scanned 35 times. For IR analysis, the rust product and tannic conversion layer were scraped off with a blade. The scraped samples and pure anhydrous KBr were mixed in the proportions of 3:100 wt.% and ground to a fine powder in a mortar with a pestle. The mixture was then pressed in a simple die to form a circular disc of about 1 mm of thickness.

The cross-section and surface morphology of the rust and tannic conversion layer were observed by scanning electron microscopy (SEM, INSPECT F, FEI, Hillsboro, OH, USA). The cross-section is perpendicular to the treated surface. For the purpose of SEM observation, the cross-sectional specimen was encapsulated into the PVC pipe filled with epoxy resin. After curing of the epoxy resin, the cross-sectional specimen was polished down to 2000 # grade emery paper and polished with terylene.

## 3. Results and Discussion

*3.1. The Influence of Concentrations of Phosphoric Acid on the Adhesion Strength*

According to Li et al. [28], the adhesion strength between the epoxy coating and the rusted steel before and after the tannic converter formula treatment were 1.93 and 5.97 MPa, respectively. When phosphoric acid was added to the tannic converter formula, the adhesion strength presented obvious changes. Figure 1 shows the variation of the adhesion strength with different concentrations of phosphoric acid. Initially, the adhesion strength was enhanced with the phosphoric acid concentration increased. However, when the phosphoric acid concentration was higher than 50 g/L, the adhesion strength reduced dramatically. In general, the formulations showed the best adhesion with 50 g/L phosphoric acid, which resulted in the adhesion strength up to 11.63 MPa. This value was much higher than the adhesion strength of samples with no treatment (1.93 MPa) or just the tannic converter formula application (5.97 MPa). The result showed that the adhesion strength was enhanced obviously with the treatment by an appropriate phosphoric acid concentration in conjunction with the tannic rust converter. It is implied that the structure and composition of the rust layer after the formulations of phosphoric acid in conjunction with the tannic rust converter treatment may play an important role on the interface strength.

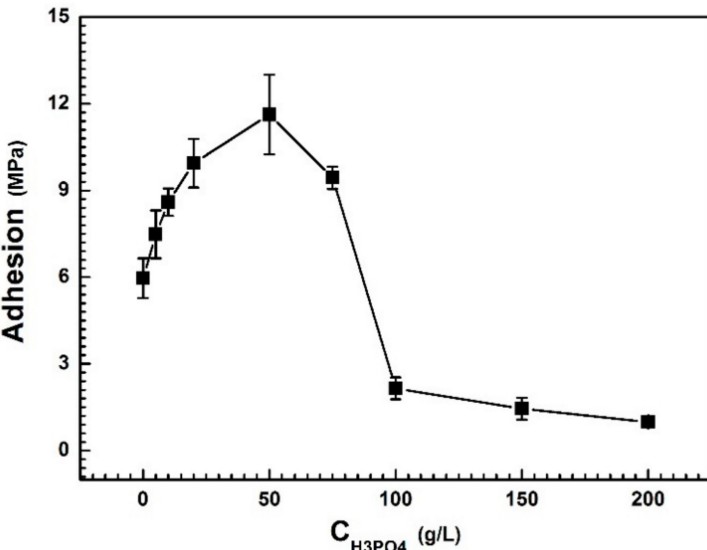

**Figure 1.** Variation of the adhesion strength with different concentrations of phosphoric acid (bars donate S.D.).

*3.2. Mechanism of Adhesion Improvement*

3.2.1. The Effect of Rust Composition

Figure 2 shows the FTIR spectra of the samples before and after treatment with 5, 50 and 200 g/L phosphoric acid in conjunction with the tannic rust converter. Raman et al. [34] suggested that the bands at 1150, 1020 and 750 cm$^{-1}$ were attributable to lepidocrocite absorptions. Among these bands the 1020 cm$^{-1}$ one is the strongest and could be considered as lepidocrocites' characteristic absorption. It has been found that that the absorption of ferric tannate located at 1340, 1207 and 1086 cm$^{-1}$ [5,35]. Gust [36] concluded that ferric phosphate showed an IR spectrum with bands located at 1044 cm$^{-1}$ (strong), 530 cm$^{-1}$ (medium). As shown in Figure 2a, the strongest absorption peak at 1020 cm$^{-1}$ showed that the main component of the original rust was lepidocrocite. After treatment with 5 g/L phosphoric acid in conjunction with the tannic rust converter (Figure 2b), the absorption peaks of lepidocrocite and ferric tannate were obvious, but it was difficult to detect that of ferric phosphate. It indicated that the conversion of the rust was not compete and the main product was ferric tannate at 5 g/L phosphoric acid, and the formation of ferric phosphate was not obvious. When phosphoric acid concentration was 50 g/L (Figure 2c), the new broad absorption peak at 1044 cm$^{-1}$ and a new absorption band appeared at 530 cm$^{-1}$ indicated the formation of the ferric phosphate, and the ferric tannate was also observed in the IR spectra. The IR results showed that phosphoric acid converted lepidocrocite into ferric phosphate and tannic acid converted lepidocrocite into ferric tannate. Though the broad and strong absorption peak of ferric phosphate at 1044 cm$^{-1}$ covered the absorption peak at 1020 cm$^{-1}$ of lepidocrocite, the absorption band at 750 cm$^{-1}$ indicated the existence of lepidocrocite. Increasing the phosphoric acid concentration to 200 g/L (Figure 2d), it detected the strong absorption peak of ferric phosphate, and the absorption bands of lepidocrocite and ferric tannate were not clear. This IR result showed that the conversion degree of the rust was further improved and most of it was converted to ferric phosphate. In general, the conversion degree of the rust was improved by the increase of phosphoric acid concentration, but above 75 g/L, the adhesion strength reduced dramatically with increase of the phosphoric acid concentration. The results showed that an appropriate conversion degree of the rust was better in consideration of adhesion strength. It is implied that the structure of the rust layer after the formulations of phosphoric acid in conjunction with the tannic rust converter treatment plays more important role on the interface strength.

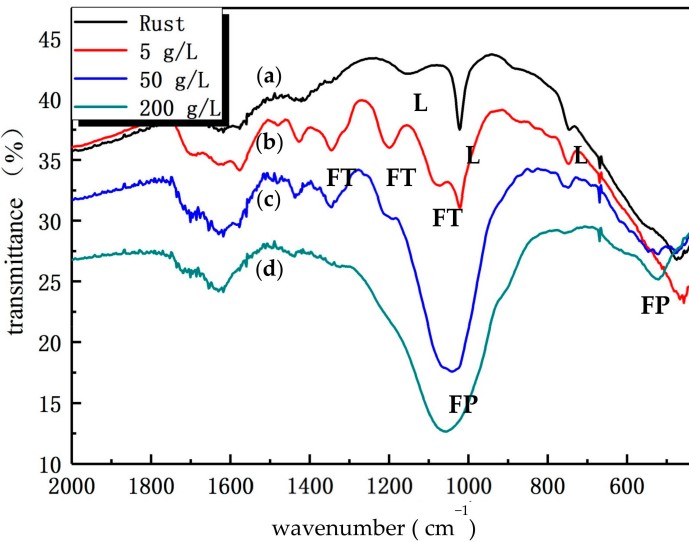

**Figure 2.** Infrared spectra for: (**a**) rust, after treatment with (**b**) 5 g/L, (**c**) 50 g/L, (**d**) 200 g/L phosphoric acid in conjunction with the tannic rust converter (L: lepidocrocite, FT: ferric tannate, FP: ferric phosphate).

### 3.2.2. The Effect of Rust Morphology

Figure 3 presents the surface and cross-section morphology of the rust layer. The rust layer presents lamellar structure with the existence of pores inside the rust. Smith and McEnaney showed that the lamellar and porous corrosion product could be lepidocrocite [37], which is in agreement with the IR result. And this porous rust layer is easily broken, which is the main reason to cause the low adhesion strength for the coating system [28].

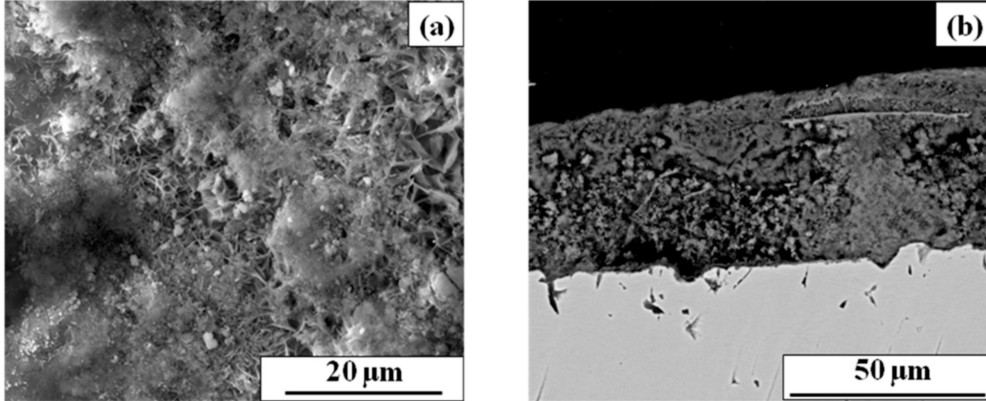

**Figure 3.** The surface (**a**) and cross-sectional (**b**) morphology of the rust layer.

After the converter treatment, the adhesion strength has dramatically changed, which could be attributed to great changes in the rust layer structure. Figure 4 presents the surface morphology of the rust layer after treatments with different concentrations of phosphoric acid in conjunction with the tannic rust converter. Compared with the morphology without treatment, the rusted structural steel represents the micro-crack structure instead of the lamellar structure. The grains became larger with the decrease of the micro-cracks between the grains by increasing the concentration of phosphoric acid. When the concentration of phosphoric acid was 5 g/L (Figure 4a), many grains of the conversion layer lifted and some of the rust particles could be seen just below the surface. By increasing the concentration of phosphoric acid to 10 g/L (Figure 4b) and 20 g/L (Figure 4c), the amount of lifted grains decreased obviously. In particular when the concentration reached 20 g/L, only a few of the grains lifted. When the concentration was 50 g/L (Figure 4d), there were no obvious

lifted grains, and the surface presented a micro-cracks structure, which was beneficial to improve the adhesion between the conversion layer and epoxy coating through the pinning effect. When the concentration of phosphoric acid reached 75 g/L (Figure 4e) and 100 g/L (Figure 4f), the grains became larger and the amount of the cracks decreased dramatically and some of the grains were completely detached. When the concentrations of phosphoric acid were 150 g/L (Figure 4g) and 200 g/L (Figure 4h), a hierarchical structure of the conversion layer were observed, together with many holes within the layer. According to Ross [16], the coating apparently penetrates into the rust through the cracks and binds the particles together. The cohesion of the coating was improved with smaller cracks between the conversion layer grains. Below 50 g/L of phosphoric acid, the conversion degree of the rust was low and the converter was not able to react with the inner rust. Though a lot of cracks were presented, the grains of the conversion layer were lifted with low adhesion strength. When the concentration was up to 75 g/L, the improvement of conversion degree of the rust resulted in the larger grains with a lower number of cracks, which reduced the epoxy coating to penetrate into the rust through the cracks and thus the adhesion strength dropped. Especially when the concentration was higher than 100 g/L, the converter reacted with the metal substrate to produce hydrogen leading to the presence of many holes and a hierarchical structure of the conversion layer. As a result, the adhesion strength decreased significantly. As described above, an appropriate size of the grains and an appropriate amount of cracks are beneficial to the adhesion improvement. And when the concentration of phosphoric acid was 50 g/L, the structure of conversion layer of best performance was obtained and the highest adhesion strength was achieved.

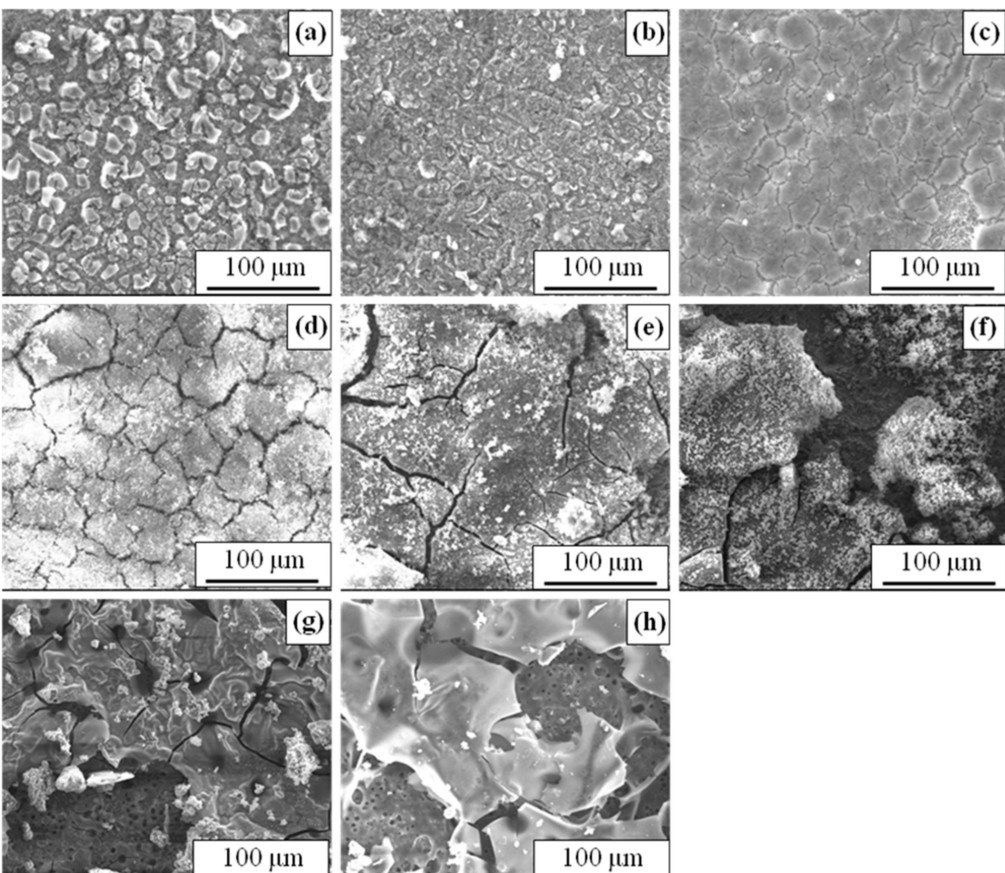

**Figure 4.** The surface morphology after 5 g/L (**a**), 10 g/L (**b**), 20 g/L (**c**), 50 g/L (**d**), 75 g/L (**e**), 100 g/L (**f**), 150 g/L (**g**), 200 g/L (**h**) phosphoric acid in conjunction with the tannic converter application.

Figure 5 presents the cross-sectional morphology and EDX results of the rust layer treated by different concentrations of phosphoric acid in conjunction with the tannic rust converter. As the element phosphorus (P) existed in the rust converter but not in the rust layer, it can indicate the penetration depth of the rust converter. When the concentration of phosphoric acid was 5 g/L, the EDX results (Figure 5b,c) showed that elemental phosphorus only existed on the surface, which indicated that the rust converter solution did not penetrate into the inner rust and react with it, leaving many holes still in the rust layer (Figure 5a).When increasing the concentration to 50 g/L, as shown in the EDX results (Figure 5e,f), the existence of P both on the surface and the bottom indicated that the rust converter had penetrated through the rust layer and reacted with it. The penetration of the rust converter and the reaction with the inner rust resulted in the holes in the rust layer disappearing instead of forming a compact conversion layer (Figure 5d). When the concentration of phosphoric acid was increased to 200 g/L, it was also found that P element was both at the surface and the bottom as shown in Figure 5h,i, which indicated that the rust converter penetrated through the rust layer. However, in this case, the cross-sectional morphology (Figure 5g) showed that the conversion layer was not compact with the existence a lot of holes and a hierarchical structure. As described above, when the phosphoric acid concentration was increased from 5 to 50 g/L, the ability of the rust converter to penetrate into the rust was improved via formation of a compact conversion layer. As a result, the adhesion strength was enhanced, but when the concentration of phosphoric acid was higher than 100 g/L, the converter reacted with the steel to produce hydrogen leading to formation of the holes as well as a hierarchical structure of the conversion layer. The hierarchical structure of the conversion layer with many holes was so incompact that it exhibited poor adhesion strength.

### 3.2.3. Adhesion Strength Failure Analysis

Figure 6 presents the structure models for the steel samples before and after rust converter treatment. Before the rust converter treatment, the structure model includes two layers (epoxy coating and rust) and two interfaces (epoxy coating/rust and rust/substrate). After the rust converter application, as a conversion layer is formed, the structure model includes three layers (epoxy coating, conversion layer and unreacted rust) and three interfaces (epoxy coating/conversion layer, conversion layer/unreacted rust, and unreacted rust/substrate).

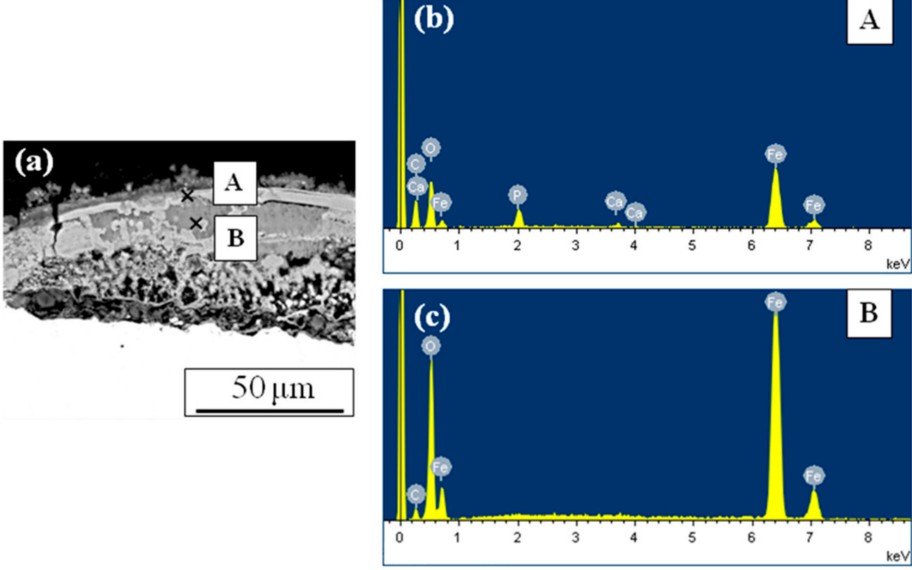

**Figure 5.** *Cont.*

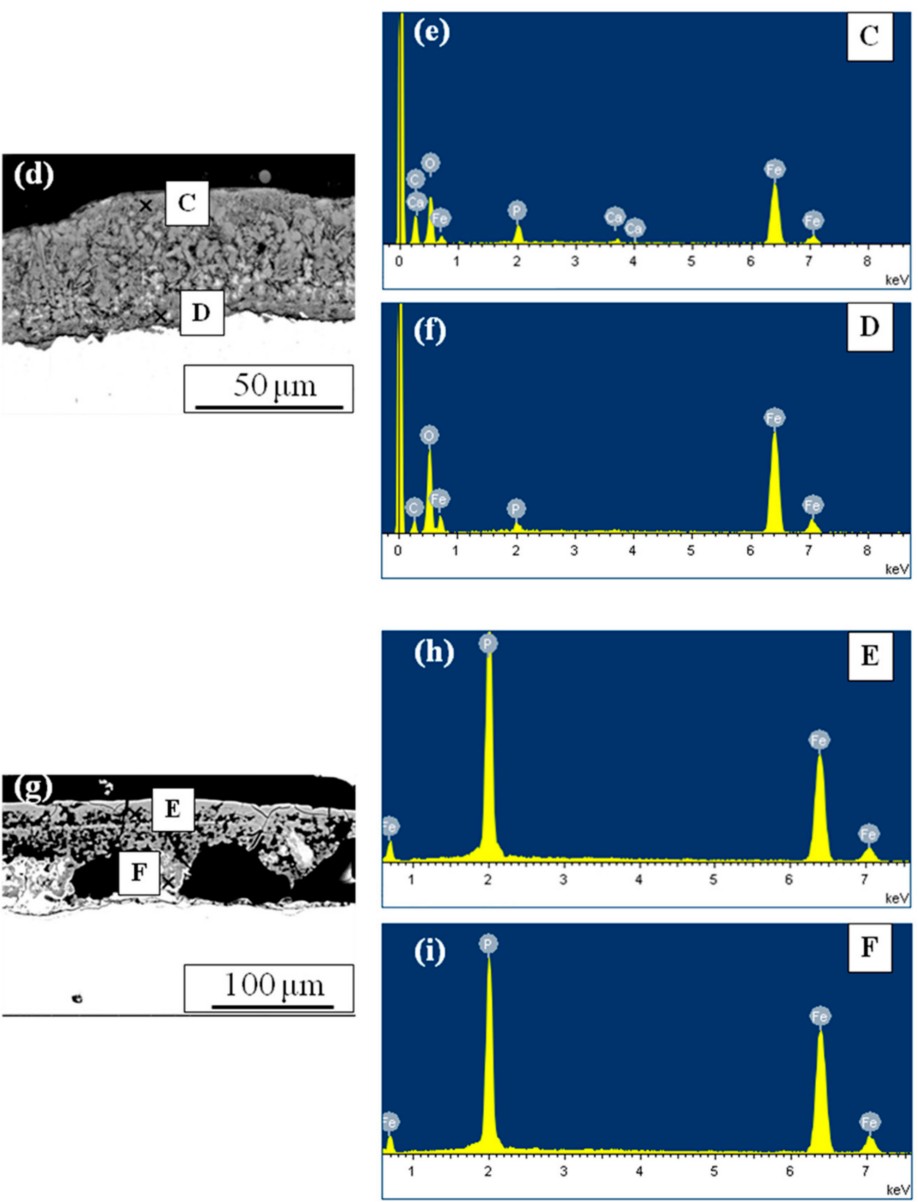

**Figure 5.** The cross-sectional morphology and EDX results of the rust treated by 5 g/L (**a–c**), 50 g/L (**d–f**), 200 g/L (**g–i**) phosphoric acid in conjunction with the tannic rust converter: the test points (**A–F**) and the relative EDX results (**b,c,e,f,h,i**)

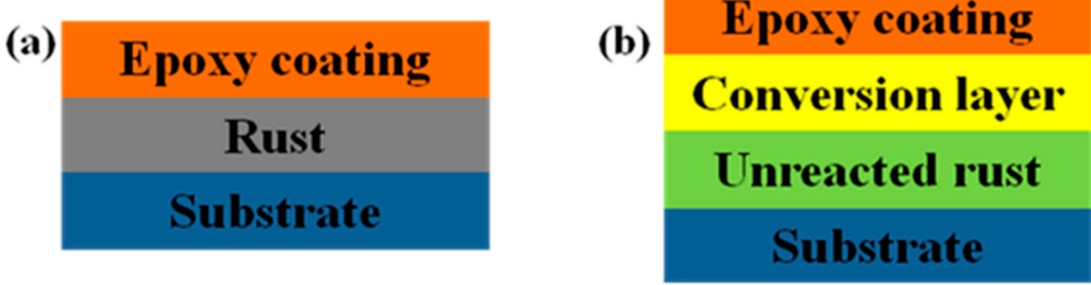

**Figure 6.** Structure models for: (**a**) before and (**b**) after rust converter treated steel samples.

Figure 7 presents the morphology after the adhesion strength measurement test with different concentrations of phosphoric acid in conjunction with the tannic rust converter application. They showed the different failure percentage in the same test area. The lowest failure percentage was observed at a phosphoric acid concentration of 50 g/L. The 5 g/L was the medium, and all the epoxy coating in the test area was detached from the substrate when the concentration was 200 g/L. It was concluded that the highest adhesion strength was obtained at an appropriate phosphoric acid concentration (50 g/L), and the high concentration (200 g/L) was adverse for the adhesion strength improvement. All of the test dollies (Figure 7b,d,f) detected some substances adhered to the epoxy coating which indicated that the interfacial bonding between the epoxy coating and the conversion layer were strong. The reason is that the residual hydroxyl groups in conversion layer may react with the active functional groups in the top epoxy coating, to form stable chemical bonds [27,38].

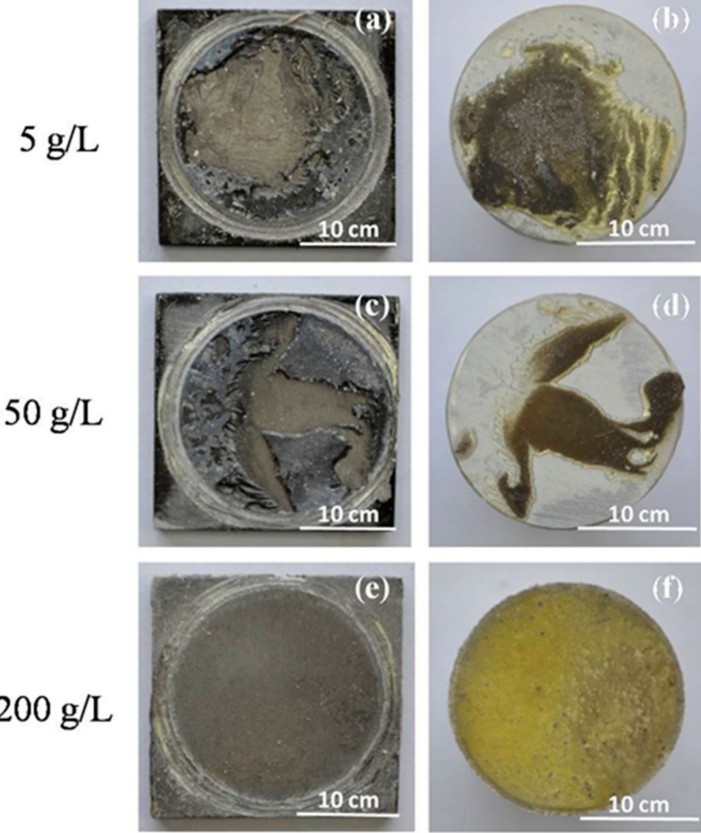

**Figure 7.** The morphology after the adhesion strength measurement test with the rust converter application: (**a**,**c**,**e**) the failure substrate; (**b**,**d**,**f**) the relative test dolly.

Figure 8 presents the surface morphology and EDX results of the conversion layer and the failure substrate treated by 5, 50 and 200 g/L phosphoric acid in conjunction with the tannic rust converter. At the concentration of 5 g/L, the micro-crack structure of the conversion layer (Figure 8a) was observed and the EDX results (Figure 8b,c) showed that the P element was present in the conversion layer. As shown in Figure 8d, no typical micro-cracks structure of the conversion layer was observed in the failure substrate. The EDX results (Figure 8e,f) showed that elemental P was also not detected in the failure substrate. This implies that the unreacted rust was the weakest structure of the coating system due to its porous structure (Figure 5a). When the concentration of phosphoric acid was 50 g/L, it showed a dense structure and it was difficult to distinguish the interface between the conversion layer and the unreacted layer as shown in Figure 5d. The surface morphology of the conversion layer (Figure 8g) and the failure substrate (Figure 8j) showed

quite different structure. The EDX results of the conversion layer (Figure 8g,i) demonstrated many times more content of elemental P than that of the failure substrate (Figure 8k,l). It indicated that the fuzzy interface between the conversion layer and the unreacted rust is the weakest structure of the coating system. When the concentration of phosphoric acid was increased to 200 g/L, the conversion layer (Figure 8m) and the failure substrate (Figure 8p) presented similar surface morphology. The content of P element between the conversion layer (Figure 8n,o) and the failure substrate (Figure 8q,r) were also very similar. It demonstrates that the conversion layer is the weakest structure of the coating system because of hierarchical and porous structure of it (Figure 5g). In general, the weakest structure of the coating system changes with the increase of the phosphoric acid concentration. The weakest structure is the unreacted rust at a low phosphoric acid concentration (5 g/L), and then changes to the fuzzy interface between the conversion layer and the unreacted rust (50 g/L). And the conversion layer becomes the weakest structure when the phosphoric acid concentration is high (200 g/L).

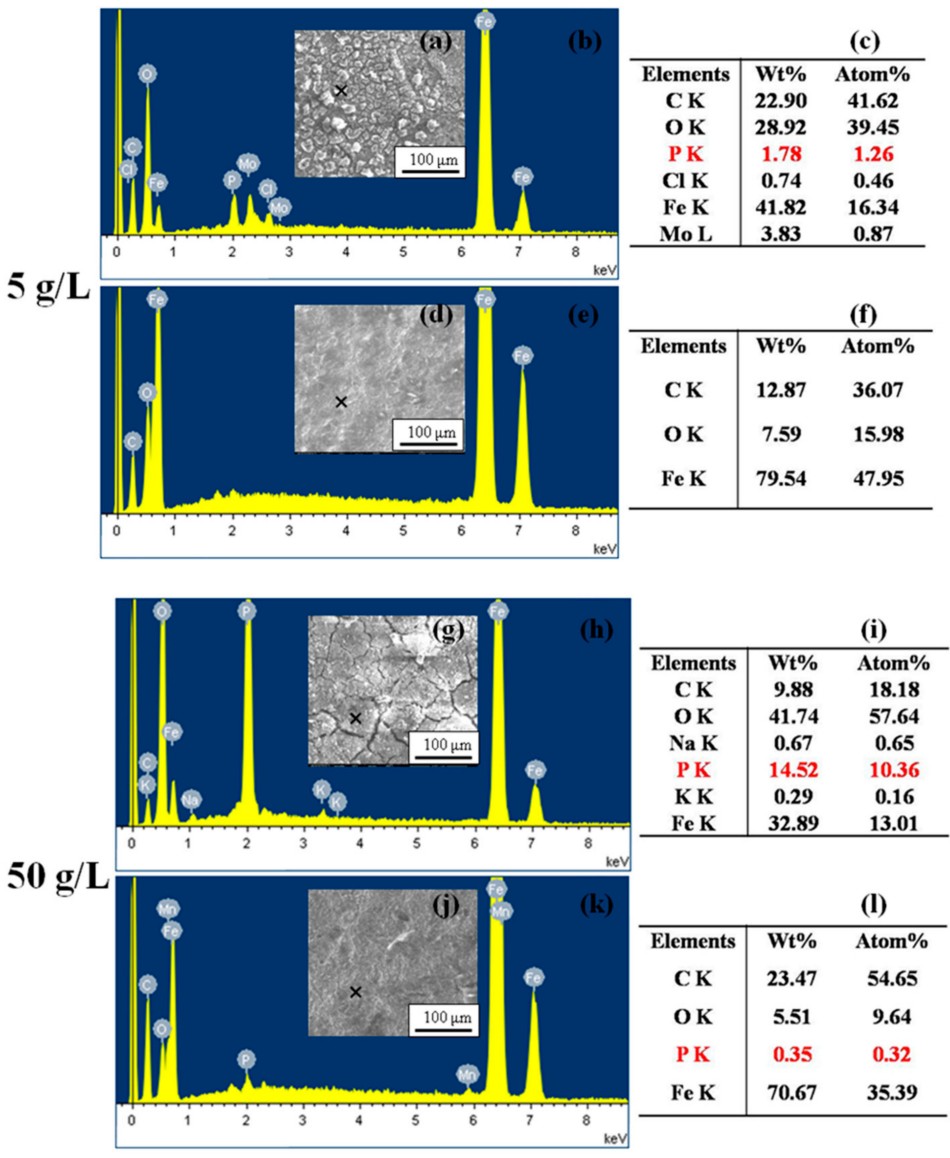

**Figure 8.** *Cont.*

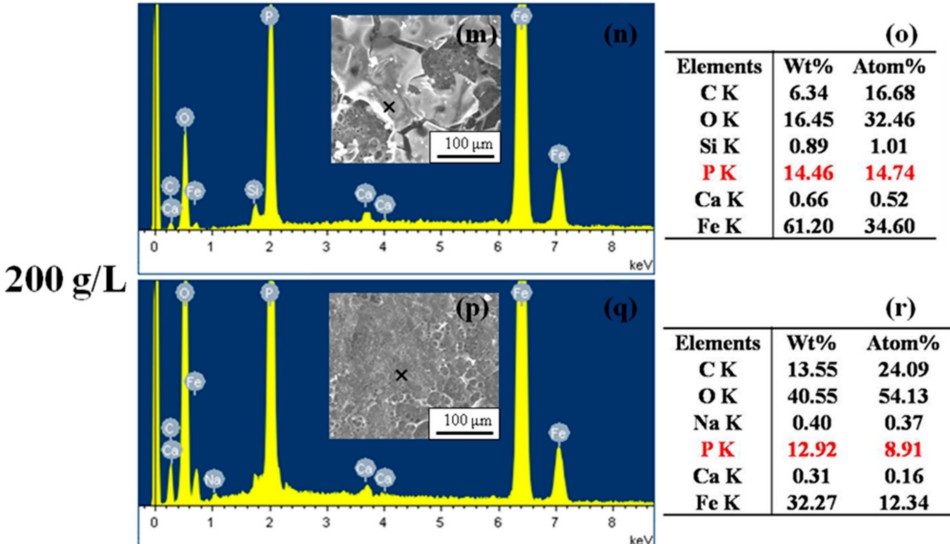

**Figure 8.** The surface morphology and EDX results of the rust treated by the rust converter: (**a**,**g**,**m**) the conversion layer, (**b**,**c**)(**h**,**i**)(**n**,**o**) the relative EDX results respectively; (**d**,**j**,**p**) the failure substrate, (**e**,**f**)(**k**,**l**)(**q**,**r**) the relative EDX results respectively.

### 3.2.4. Influence of pH and Phosphate Radical on Adhesion Strength

Figure 9 presents the influence of pH and phosphate radical on adhesion strength. After adding 50 g/L phosphoric acid to the tannic rust converter, the pH value changed from 2.58 to 0.38 and the obtained highest adhesion strength was 11.63 MPa. When adjusting the pH value by HCl to 0.38 without adding the phosphoric acid, the adhesion strength dropped to 4.47 MPa. When 50 g/L phosphoric acid mixed in the tannic rust converter and adjusted the pH value by NaOH to 2.58, the adhesion strength was 6.72 MPa which was much lower than that without adjusting the pH value. These results show that both of the pH and phosphate radical are the important factors to improve the adhesion strength.

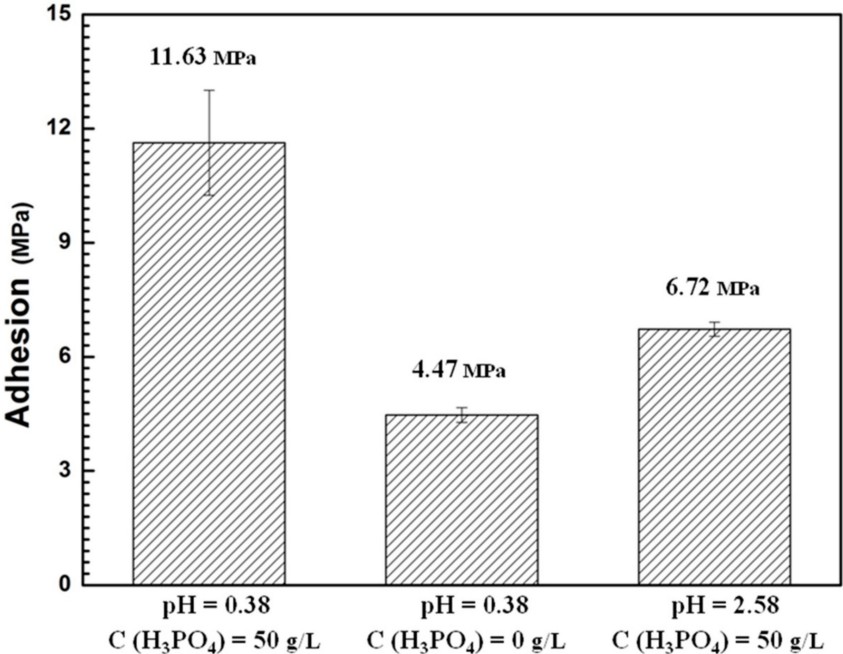

**Figure 9.** Influence of adhesion strength by pH and phosphate radical (bars donate S.D.).

It could be concluded that the combination of tannic and phosphoric acid seemed to create a competing reaction between them when applied to lepidocrocite. Phosphoric acid seemed to dominate regardless of the tannic acid concentration [6]. According to Rahim [27], the ferric phosphate precipitation reaction is as shown in Equation (1). Therefore, after adding 50 g/L phosphoric acid to the tannic converter and adjusting the pH value to 2.58, the dominate reaction was the phosphoric acid with lepidocrocite, and the increase of the pH increased the deposition rate of the ferric phosphate [4]. The high deposition rate of the ferric phosphate might lead to fast formation of the large grains of ferric phosphate conversion layer on the surface of the unreacted rust (Figure 10a), which influenced the penetration of the rust converter solution, and the inner rust might not react with the rust converter sufficiently. Therefore, most of the inner rust was not converted and presented porous structure (Figure 10b). On the other hand, the fast formation of conversion layer might lead to the uncomplete release the stress, then the stratification structure of the conversion layer presented, the surface and cross-sectional morphology both proved it. These two reasons resulted in low adhesion strength.

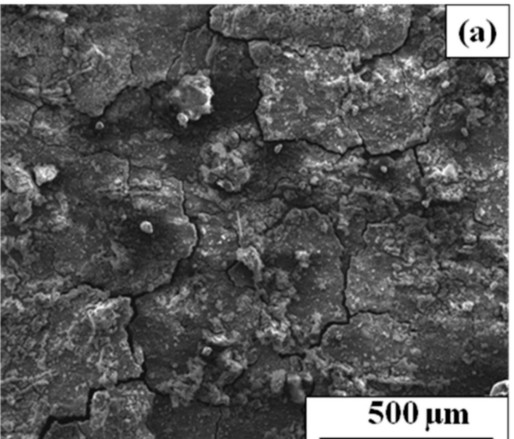 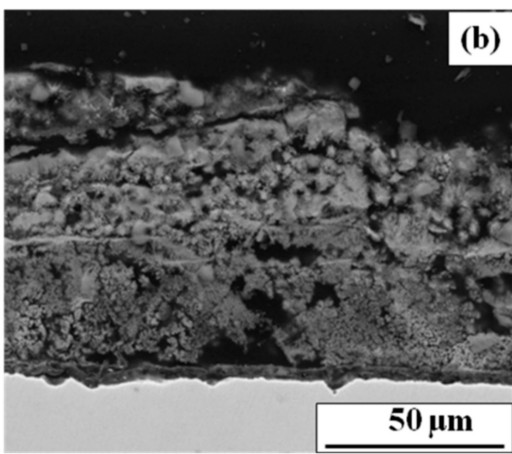

**Figure 10.** The surface (**a**) and cross-sectional (**b**) morphology of the samples with 50 g/L phosphoric acid mixed in the tannic rust converter application (adjusting the pH value to 2.58 by NaOH).

Martin [39] pointed out that the rust showed high dissolution rate in acidic pH, as the main component of the rust was lepidocrocite ($\gamma$-FeOOH), and it could provide sufficient ferric ion while the converter solution was applied (Equation (2)). The ionization of tannin was shown in Equation (3), and according to Ross et al. [16], three tannate ions reacted with each ferric ion to form a stable octahedral coordination compound (Equation (3), R represents the rest of the tannin molecule). When adjusting the pH value by HCl to 0.38 without adding the phosphoric acid, the high acidic pH would prevent the formation of tannate ions, which reduced the conversion efficiency of the tannin. Li et al. [28] suggested that the degree of conversion with the rust was the key issue for the further adhesion improvement of the coating system. The high acidic pH decreased the degree of conversion with the rust. As shown in Figure 11a, lifted grains were formed at the conversion layer and the rust particles were just below it. On the other hand, the reduced conversion efficiency of the tannin could not react with the inner rust sufficiently and left the porous rust layer (Figure 11b), which showed low adhesion strength.

$$H_3PO_4 + \gamma\text{-FeOOH} \rightarrow FePO_4\downarrow + 2H_2O \tag{1}$$

$$\gamma\text{-FeOOH} + 3H^+ \rightarrow Fe^{3+} + 2H_2O \tag{2}$$

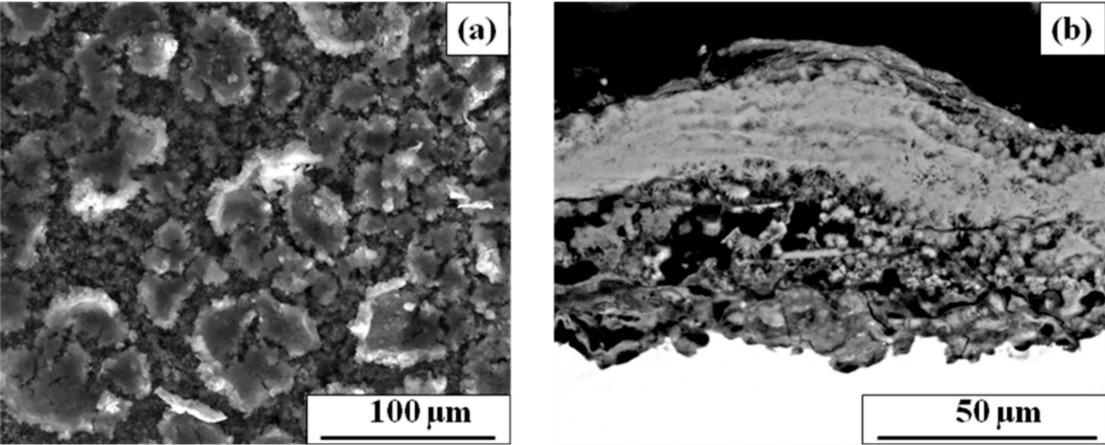

**Figure 11.** The surface (**a**) and cross-sectional (**b**) morphology of the samples without phosphoric acid mixed in the tannic rust converter application (adjusting the pH value to 0.38 by HCl).

$$\qquad\qquad\qquad\qquad\qquad\qquad\qquad\qquad\qquad\qquad\qquad (3)$$

$$\qquad\qquad\qquad\qquad\qquad\qquad\qquad\qquad\qquad\qquad\qquad (4)$$

As discussed above, the combination of the pH and the phosphate radical played far better enhancement of the adhesion strength of the coating system than applying either of them separately.

## 4. Conclusions

The influence of phosphoric acid in conjunction with a tannic converter formula on the adhesion of a coating system was evaluated by employing an epoxy coating as the top coating. The highest adhesion strength (11.63 MPa) was obtained when the concentration of phosphoric acid was 50 g/L, which was far better than the treatment without phosphoric acid (5.97 MPa) or no rust converter treatment (1.93 MPa). Furthermore, the adhesion was close to the value presented in literature for epoxy and steel (13.14 MPa) [40]. The improvement of the adhesion strength by the phosphoric acid was attributed to the formation of the compact and micro-crack conversion layer. The porous rust reacted with the converter and the compactness was improved obviously, which enhanced the cohesion of the rust layer. This improvement of cohesion of the rust layer increased the adhesion strength of the coating system. On the other hand the coating apparently penetrated into the inner conversion layer through the cracks and bound them together. Consequently, the adhesion between the conversion layer and the coating was improved. These two reasons led to a significant increase in the adhesion of the coating system. It was found that both of the pH and phosphate radical were the important factors to determine the adhesion strength, and the combination of the pH and the phosphate radical resulted in a far better enhancement of the adhesion strength of the coating system than applying either of them alone.

**Author Contributions:** Conceptualization, Y.L.; methodology, Y.L. and B.L.; validation, X.G.; formal analysis, Y.L. and B.L.; investigation, X.G.; resources, B.L.; data curation, Y.L.; writing—original draft preparation, Y.L.; writing—review and editing, X.G.; visualization, B.L.; supervision, X.G.; project administration, Y.L.; funding acquisition, B.L. All authors have read and agreed to the published version of the manuscript.

**Funding:** This research was funded by the Fundamental Research Funds for the Central Universities (Grant No. 20lgpy75).

**Institutional Review Board Statement:** Not applicable.

**Informed Consent Statement:** Not applicable.

**Data Availability Statement:** The data presented in this study are available.

**Conflicts of Interest:** The authors declare no conflict of interest.

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
