# Peer review of "Influence of Phosphoric Acid on the Adhesion Strength between Rusted Steel and Epoxy Coating"

_coatings, doi:10.3390/coatings11020246_

Round 1

Reviewer 1 Report

The article "Influence of phosphoric acid on the adhesion strength between rusted steel and epoxy coating" presents interesting research on increasing the adhesion of protective coatings for steel surfaces. The topic is interesting especially for practical applications and good described.

1. In Chapter 1, the review of literary research is modest and the justify of presented research is insufficient. On the other hand, in chapter 3 the results of the literature research were posted with their own results, which creates some confusion. It may be consideration to re-edit the text by transferring the results of the literature research to the first chapter and providing better  justification for take of the such research.

Thanks for the reviewer’s suggestion and we have re-edited the introduction in the revised paper. (line 43-48,59-69).

2. In Figure 2, the y axis should be described.

The description of y axis has been added in Figure 2. (line 185).

3. Figure 3, 4, 10 ,11 should provide additional explanations of these structures.

The reviewer’s suggestion is considered and the relevant additional explanations of the structures have been indicated in the revised paper. (line 193-195, 209-213, 360-365, 382-385).

4. In Figure 5 there are additional symbols (a, B, C, - ..) which are not explained in the caption.

In the revised paper, according to the reviewer’s suggestion, all the additional symbols have been explained in the caption,and added “the test points (A,B,C,D,E,F) and the relative EDX results (b,c,e,f,h,i)”. (line 273-274).

5. The similar remark applies the Figure 8.

In the revised paper, all the additional symbols have been explained in the caption, and added “(a) (g) (m) the conversion layer,(b, c)(h, i)(n, o) the relative EDX results respectively; (d) (j) (p) the failure substrate, (e, f)(k, l)(q, r) the relative EDX results respectively”. (line 336-337).

6. The conclusions could be reformulated to better present the obtained research results.

We would like to thank the reviewer’s helpful advice and have modified the conclusion in the revised paper. (line 396-406).

7. In the references, part of the articles can be supplemented with DOI or website address.

In the references, we have supplemented with DOI or website address in the revised paper. (lines 420-504).

8. On line 103, symbol Φ may be changed.

We have deleted the Φ,and added “diameter” instead in the revised paper. (line 107-108).

Reviewer 2 Report

The paper of Yang Li et al is related for the system of rusted steel and epoxy coating preparation and characterization. The topic is generally interesting, however the paper contains unexplained places (below) and need major revisions.

1. In 2.1 Materials, please explain, while the sample for corrosion was exposed for 72h exactly in the salt spray chamber and which impact it can have on obtained results?

The salt spray test is generally accepted as a common method to accelerate corrosion. The purposes of the sample for corrosion exposed for 72h in the salt spray chamber were to obtain thick rust layer quickly. The results showed that the rust layer formed was porous structure (as shown in Figure 3), and the main component was lepidocrocite (as shown in Figure 2), which was in agreement with the authors’ study [1]. The porous rust layer was easily broken, which was the weakest link to cause the low adhesion strength for the epoxy coating system [2].

2. In Fig. 9 it shown the impact of pH on adhesion stregth. However, what is the impact of pH on the chemical rust composition? Can You show the corresponding IR spectra? What can be the impact of NaOH on adhesion strength, page 11, line 321?

Thanks for the reviewer’s comment.

(1)As shown in Figure 2 (c) , when the concentration of phosphoric acid was 50 g/L, the IR spectra result showed that the composition of the conversion layer contained lepidocrocite, ferric phosphate and ferric tannate. Several authors’study [3] showed that the rust presented high dissolution rate with the increase of the acidic pH, and there was no evidence that the pH changed the chemical rust composition. Therefore, when 50 g/L phosphoric acid was mixed in the tannic rust converter application and the pH value was adjusted to 2.58 by NaOH, we might speculate that the composition of the conversion layer still contained lepidocrocite, the ferric phosphate and the ferric tannate. At the same time, when no phosphoric acid was mixed in the tannic rust converter application and the pH value was adjusted to 0.38 by HCl, we might speculate that the composition of the conversion layer contained lepidocrocite and ferric tannate.

(2)As discussed above, we have studied the effect of pH on adhesion strength, and the NaOH is the reagent to adjust the pH.

3. SEM micrographs should have a profesional scale bar markers. At the moment it is not possible to read the description of scale in Fig.4. Also beeter to use the same scale for all SEM images for comparison.

We would like to thank the reviewer’s helpful advice and the SEM micrographs have been modified in the revised paper. Most of time, we follow the rules to use the same scale for the SEM images for comparison. Sometimes for the sake of better presenting the structural characteristics,different scales of the surface and cross-section SEM images might be used.(line 198, 235-236, 266-271, 329-334, 365-368).

4. The impact of phosphate radicals on the adhesion strength should be more explained. What is the relation between phosphate radicals concentration and the the adhesion strength?

Thanks for the reviewer’s suggestion. We have added more explanations about the impact of phosphate radicals on the adhesion strength in the revised paper. In addition, the relevant description has been indicated in the revised paper (line 360-365).

The chemical equation (Fe3+ + PO43- = FePO4  ) showed that with increase of the concentration of phosphate radicals, the reaction rate of phosphate and iron ions should be faster , resulting in faster deposition rate of the conversion layer. The grains were formed very large under the rapid deposition condition, which hindered the further penetration of the conversion liquid into the inner rust layer to react with it, resulting in the retention of the porous inner rust layer. On the other hand, due to the internal stress could not be fully released by rapid deposition, the conversion layer was stratified. The two aspects indicated that simply increasing the phosphate concentration would greatly reduce the adhesion of the coating system, and the experimental results also proved this point.

5. In conclusions, please compare your obtained best value of the adhesion with best values presented in literature for epoxy and steel.

Thank you for the reviewer’s comments . We have searched literature and compared our best value of the adhesion with best values presented in literature for epoxy and steel in the revised paper. (line 396-397).

6. English need minor revisions.

Per your suggestion, we have modified part of the expression carefully in the revised paper. (line 53, 55, 73-76, 123-124, 178, 180-181, 191-195, 209-213, et al.)

References

  1. Génin J.-M. R., Refait Ph. & Abdelmoula M.Green Rusts and Their Relationship to Iron Corrosion; a Key Role in Microbially Influenced Corrosion. Hyperfine Interact., 2002, 139, 119-131. https://link.springer.com/article/10.1023/A:1021219021919
  2. Li, ; Ma, Y.T.; Zhang, B.; Lei, B.; Li, Y. Enhancement the adhesion between epoxy coating and rusted structural steel by tannic acid treatment. Acta Metall. Sin. (Engl. Lett.) 2014, 27, 1105–1113. https://link.springer.com/article/10.1007/s40195-014-0132-5
  3. Martin, T. Precipitation and Dissolution of Iron and Manganese Oxides. Environ. Catal. 2005, 1, 61–82. DOI: 10.1201/9781420027679

Round 2

Reviewer 1 Report

Additional explanations in figure captions are intended to help readers who are only viewing the article to find out more about its content. However, the captions are still very modest and there is no complete information about the structures presented in the figures. 

Reviewer 2 Report

Authors make proper corrections according to referee remarks and I suggest publish the paper as it is.